# OpenReview forum: "Localize and Neutralize: Gradient-Guided Token Suppression Against Visual Prompt Injection Attack"
_ICML.cc/2026/Conference — ICML 2026 regular_

### Official Review · Reviewer_cr7C · 2026-03-06

**Soundness:** 3
**Presentation:** 3
**Significance:** 3
**Originality:** 3
**Overall Recommendation:** 4
**Confidence:** 3

**Summary:**

This paper studies defense against perturbation-based visual prompt injection attacks on LVLMs. The key observation is that attack effectiveness is concentrated on a sparse subset of visual tokens rather than spread across the whole image. Based on this, the paper proposes GTM, which computes a hidden-state-gradient saliency score for image tokens, masks the top-k critical tokens, and then runs generation on the sanitized embeddings. The method is inference-time, training-free, and only needs one partial forward-backward pass. Experiments on general prompt injection and jailbreak settings show strong ASR reduction, while largely preserving benign-task utility and adding little runtime overhead.

**Compliance With Llm Reviewing Policy:**

Affirmed.

**Key Questions For Authors:**

1. How sensitive is performance to the masking budget across different models and attack types? Whether this choice transfers cleanly or needs retuning per model.

2. The paper argues that hidden-state gradients better handle token alignment. Could the authors provide a more direct empirical analysis of how often aligned vs. misaligned cases occur in practice, and how much this matters for overall defense performance?

**Limitations:**

Limitations and potential negative impacts have been sufficiently discussed in the paper.

**Strengths And Weaknesses:**

**Strengths**

1. The paper goes beyond jailbreak-only defenses and targets more general visual prompt injection, which is a real gap in the current defense literature.

2. The core idea is simple and fairly convincing. The masking analysis gives a clean story: only a small subset of image tokens seems to matter for attack success, so localize them and neutralize them. This makes the defense easy to understand.

3. The method performs well across VMA, ImgHijack, and jailbreak settings, and is competitive with or better than strong baselines.

**Weaknesses**

1. The arguments rely on token alignment, small perturbation, and local smoothness/linearity assumptions. These assumptions help motivate the method, but it is still unclear how much of the theory really carries over to modern LVLMs in practice.

2. The paper is mainly about perturbation-based attacks. It is less clear whether the same defense idea would help against structure-based visual prompt injection, visible text attacks, or mixed attacks. The related-work section explicitly separates these settings, but the experiments stay on the perturbation side.

3. In the aligned case, the proposed attribution improves over logit-based attribution, but the gap is not dramatic. This makes the empirical payoff of the theoretical section look a bit modest.

---

> ### Author Rebuttal · Authors · 2026-03-31
>
> We greatly appreciate your responsible and meticulous review. Your valuable feedback will improve our work greatly.
> >**W1. The Practicality of Assumption**
>
> 1. **Token Alignment as a Realistic Threat**: Token alignment represents a highly realistic and challenging attack scenario where an adversary deliberately aligns the first target token with a model's natural prediction to stealthily hijack the generation process.
> 2. **Validity of Small Perturbation and Linearity**: The assumptions of small perturbations and local linearity are highly reasonable in this context. Adversarial attacks explicitly bound their noise (e.g., under strict $l_{\infty}$ constraints) to ensure the perturbations remain visually imperceptible to humans. Within these tightly constrained perturbation spaces, local linear approximations—like those utilized in our gradient-based attribution—have been extensively validated and proven empirically robust across the broader field of adversarial machine learning.
>
> >**W2. Defense Scope**
>
> We respectfully clarify that our research is specifically targeted at perturbation-based prompt injection attacks, a scope that was explicitly defined in the original manuscript. As noted in the Summary of Contributions (Lines 103–105), GTM is designed as an "efficient and principled defense for perturbation-based visual prompt injection attacks." This focus is further reflected in our empirical evaluations, as stated in the Introduction (Lines 087–089), and is consistently reiterated in the Experimental Setup (Line 266).
>
> Concretely, GTM is not well-suited to defending against structure-based prompt injection attacks, and the underlying reason lies in their gradient behavior. Structure-based attacks, such as those that explicitly render malicious text within the image, produce visual features that are highly similar to those of natural images. As a result, their gradient distributions closely resemble those of benign inputs and do not exhibit the extreme token sparsity that is characteristic of perturbation-based attacks. Therefore, it is difficult to distinguish such structural triggers from normal visual context using gradient-based features alone.
>
> We thank the reviewer for pointing this out. We will explicitly clarify the scope of applicability in the revised manuscript and include a detailed discussion of limitations.
>
> >**Q1. Sensitivity to Mask Budget**
>
> As shown in our main exexperiments, **a single default masking budget(mask ratio) already delivers consistently strong performance across different LVLMs and a wide range of attack types**, without the need for per-model tuning. Besides, our ablation results suggest that GTM is not particularly sensitive to this parameter. Varying the masking ratio leads to a gradual and predictable safety–utility trade-off, rather than sharp performance degradation. In practice, this means the same setting transfers well across models, while still leaving room for lightweight adjustment if a specific deployment scenario calls for it.
>
> >**W3&Q2. Practicality of Token Alignment**
>
> Token alignment is not merely a theoretical edge case; it has become an increasingly deliberate and effective strategy in state-of-the-art multimodal jailbreak attacks to bypass safety filters. Recent advanced attacks [1][2] explicitly leverage techniques such as textual steering or prefix injection to anchor the model’s initial generation phase. By forcing an aligned starting token, they can obscure adversarial intent and thereby significantly improve the attack success rate (ASR).
>
> We conducted additional ablation studies on [1] and [2], using Full Advbench as the evaluation dataset and Qwen2-VL-7B as the target model. The ASR(%) results are as follows:
>
> | Attack Method | W/O Defense | Logit-based | Ours  |
> | :--- | :--- | :--- | :--- |
> | UltraBreak (Cui et al.,2026) | 71.54 | 13.08 | **7.88** |
> | JPS (Chen et al., 2025) | 97.69 | 31.54 | **20.19** |
>
> It can be observed that our proposed method significantly improves the defense success rate under token alignment scenarios.
>
> [1] Cui, K. et al. (2026). Toward Universal and Transferable Jailbreak Attacks on Vision-Language Models. In  International Conference on Learning Representations (ICLR).
> [2] Chen, R. et al. (2025). JPS: Jailbreak Multimodal Large Language Models with Collaborative Visual Perturbation and Textual Steering. In  ACM Conference on Multimedia (ACM MM).

---

> > ### Author Rebuttal · Reviewer_cr7C · 2026-04-04
> >
> > Thank you for the response. My concerns have been adequately addressed.

---

> > > ### Author Response · Authors · 2026-04-05
> > >
> > > Thank you very much for your valuable feedback and kind support. We are pleased to know that the additional experiments have resolved your concerns. Your recognition of our work and your recommendation for acceptance are truly encouraging and deeply appreciated.
> > >
> > > As mentioned, we will continue to refine both the main manuscript and supplementary materials in the revised version, incorporating further experiments and improvements where needed.

---

### Official Review · Reviewer_Yf8y · 2026-03-06

**Soundness:** 2
**Presentation:** 2
**Significance:** 3
**Originality:** 3
**Overall Recommendation:** 2
**Confidence:** 3

**Summary:**

This paper proposes Gradient-guided Token Masking (GTM), motivated by the observation that perturbation-based visual prompt injection attacks rely on a small subset of critical image embedding tokens rather than uniformly on all tokens. GTM localizes these critical tokens using gradient-based attribution and then neutralizes the attack by masking (zeroing out) the top-ranked tokens in the image embedding sequence. A key technical point is that attribution based on the output probability gradient of the first generated token can fail under token-alignment scenarios, where the first adversarial target token matches the model’s natural first token. To address this, GTM uses a hidden-state–based attribution score (the gradient norm of the first-token hidden-state norm with respect to each image token), and provides theoretical justification that this score preserves the token-importance ranking of the full adversarial loss gradient up to scaling under mild assumptions. The method is computationally efficient, requiring only a single partial forward–backward pass to compute token scores and apply masking. Experiments across multiple LVLMs, diverse prompt injection and jailbreak attacks, and different perturbation budgets show that GTM significantly reduces attack success rates while largely preserving utility on benign tasks, with minimal runtime and memory overhead compared to heavier defenses.

**Compliance With Llm Reviewing Policy:**

Affirmed.

**Final Justification:**

I sincerely thank the authors for the rebuttal. However, the authors did not provide a reasonable explanation for why the stronger adaptive attacks (Strategy 1 and Strategy 2) are not effective against the proposed defense. They only explain why the Strategy 3 adaptive attack does not work, stating that its loss does not decrease. Therefore, I maintain my score.

**Key Questions For Authors:**

- Can you evaluate GTM against additional inference-time defenses not included in the current comparison, such as more general purification or transformation-based defenses, to better contextualize its performance and trade-offs?

- How robust is GTM in an adaptive-attacker setting where the attacker knows the defense (Kerckhoffs’s principle)? For example, if the attacker explicitly regularizes the perturbation to avoid producing a small set of high-saliency tokens (i.e., spreads the signal across more tokens/pixels), how does GTM perform?

- Do you have an explanation for why perturbation-based visual prompt injection attacks rely on only a small subset of critical tokens? Can you provide additional analysis connecting this sparsity to prior observations about adversarial perturbations concentrating on semantically salient regions (e.g., edges/objects) or to related “semantic gradient” interpretations?

**Limitations:**

yes.

**Strengths And Weaknesses:**

# Strengths

- The proposed method is computationally efficient and practical, since many defenses (e.g., adversarial training) can add substantial overhead.
- The algorithm box is clear and makes the method easy to understand and reproduce.
- The paper provides a principled motivation (critical-token sparsity) and supports it with both empirical analysis (masking-based study) and theoretical justification for the proposed attribution score.
- The experimental evaluation is broad, covering multiple LVLMs and multiple attack settings (general prompt injection and jailbreak), with comparisons to several baselines.
- The paper explicitly evaluates the utility/accuracy impact on benign tasks and reports efficiency metrics (runtime and memory), which helps assess deployability rather than only reporting attack success rates.

# Weaknesses

- The paper does not evaluate against some common inference-time defenses outside the compared set (e.g., more general purification or transformation-based defenses), which would help contextualize where GTM stands relative to alternative deployment options.
- The evaluation does not fully address the adaptive-attacker setting where the defense mechanism is known to the attacker (Kerckhoffs’s principle). An attacker might try to distribute the adversarial signal across more tokens/pixels to avoid producing a small set of high-saliency tokens that are easy to mask.
- Figure 4 seems oversized relative to the amount of information it conveys. It could likely be made smaller, merged with related results (e.g., Table 6), or converted into a compact table/combined figure for readability.
- The paper does not explain why the attacks rely on only a few critical tokens. This behavior might relate to prior observations that adversarial signals tend to concentrate on semantically salient regions (e.g., edges or objects) and connect to “semantic gradient”–style interpretations discussed in [1],[2],[3].

[1] Pang, T. et al. (2022). Robustness and accuracy could be reconcilable by (proper) definition. In International Conference on Machine Learning (ICML).

[2] Tsipras, D. et al. (2019). Robustness may be at odds with accuracy. In International Conference on Learning Representations (ICLR).

[3] Jung, J. et al. (2024). PeerAiD: Improving adversarial distillation from a specialized peer tutor. In Proceedings of the IEEE/CVF Conference on Computer Vision and Pattern Recognition (CVPR).

---

> ### Author Rebuttal · Authors · 2026-03-31
>
> Thank you for your valuable feedback, which will enhance the integrity of our paper. To address your concerns, we have provided additional experiments. We sincerely hope our response resolves the concerns raised, and we would greatly appreciate reconsideration of the score.
>
> >**W2&Q2. Adpative Attack**
>
> Evaluating robustness under adaptive and worst-case attackers is indeed crucial, and we agree with the reviewer that an adaptive adversary may attempt to disperse adversarial signals to bypass our masking strategy. Following Kerckhoffs’s principle and the six principles outlined in [1], we have rigorously devised an adaptive attack framework.  We design an adaptive attack based on **random masking**, where 50% of the image embedding tokens are stochastically masked during optimization. This encourages the attacker to learn redundant and spatially distributed adversarial features. Importantly, to properly optimize under such stochastic transformations, we adopt an **Expectation over Transformation (EOT)**[2] strategy during attack generation，which is a widely recognized adaptive attack method. We trained for 10,000 epochs. The ASR results are summarized below:
>
> | | Without Masking | Random Masking|
> | :--- | :---: | :---: |
> | **GTM** |0.00|4.55|
>
> As shown, even under this strengthened adaptive attack with EOT optimization, the attack fails to effectively bypass our defense. This demonstrates that GTM maintains strong robustness against adaptive strategies that attempt to distribute adversarial perturbations across tokens.
>
> Should the reviewer have any further suggestions regarding the design of the adaptive attack, please let us know. We are fully committed to incorporating additional experiments to address your concerns.
>
> >**W1&Q1. Transformation-based Defenses**
>
> First, we would like to politely direct the reviewer to our existing evaluations against DiffPure, a powerful diffusion-based purification defense. Second, we compared our proposed method against two transformation-based baselines: **R&P** and **JPEG compression**. We conducted **adaptive attacks** under the same attack budget and examined their utility preservation on the MM-Vet benchmark. The adaptive attack methods tailored for JPEG compression and Randomization (RP) are derived from [2]. The results are presented in the table below:
>
> | Method | Adaptive Attack ASR (%) | MM-Vet  |
> | :--- | :---: | :---: |
> | R&P | 55.45 | 58.5 |
> | JPEG Compression | 35.45 | 60.6 |
> | **GTM** | 4.55 | 56.8 |
>
> As demonstrated in the results, while our method achieves comparable utility preservation to these baselines, it exhibits significantly greater robustness against adaptive attacks.
>
> >**W3. Paper Formatting Issues**
>
> We sincerely thank the reviewer for their insightful and detailed suggestions. We will carefully address these points in the next revision to ensure that the final version incorporates more comprehensive and essential data.
>
> >**W4&Q3. Semantic Gradient Visualization**
>
> We conducted the following experiment for analysis: First, for a given adversarial image, we performed a forward pass to compute the model's initial output, then used the $l2$-norm gradient to assess the adversarial influence on the input image. Next, we employed the Gradient Token Masking (GTM) defense mechanism to truncate the gradients of all tokens, retaining only those identified as critical, and backpropagated these gradients to the original image. For the tokens not selected, we repeated this process to ensure comprehensive coverage and identify all tokens that could impact the success of the attack. Finally, we generated a gradient heatmap to visually display the most influential tokens in the adversarial attack. The results are shown in https://anonymous.4open.science/r/ICML-F31882/f1.png.
>
> **We found that the gradients formed by the adversarial tokens were not concentrated on "feature areas"** such as the dog's core face or torso. **Instead, they exhibited a highly structured, horizontal band-like distribution with a strong spatial location preference or structural anomaly**. This suggests that the concentration of tokens in visual prompt injection attacks targeting LVLMs is driven by factors distinct from the specific distribution of adversarial noise formed in traditional classifiers. The exact mechanism behind this remains unexplored. We hypothesize that this distinctive gradient distribution and its concentration may be related to the positional encoding mechanism in the Transformer architecture. The underlying causes, however, remain to be systematically investigated in future work.
>
> [1] F., Tramer  et al. (2020). On Adaptive Attacks to Adversarial Example Defenses. In Advances in Neural Information Processing Systems (NeurIPS). [2] A. Athalye et al.(2018) Obfuscated Gradients Give a False Sense of Security: Circumventing Defenses to Adversarial Examples. In International Conference on Machine Learning (ICML).

---

> > ### Author Rebuttal · Reviewer_Yf8y · 2026-03-31
> >
> > I thank the authors for their detailed rebuttal and helpful clarifications. Regarding the adaptive attack experiment, I still find the evaluation somewhat unclear. In particular, it is not clear which data and model were used. Random masking also appears to be a relatively weak adaptive attack because it does not directly optimize against the proposed defense mechanism. A stronger adaptive evaluation would be a defense-aware gradient-based attack that explicitly optimizes attack success after token masking or regularizes the saliency distribution to avoid concentrating adversarial influence on only a few tokens. Such an experiment would make the robustness claim more convincing.

---

> > > ### Author Response · Authors · 2026-04-05
> > >
> > > We sincerely appreciate your patient and constructive feedback, which will greatly assist us in further refining our manuscript.
> > >
> > > >**Q1. Details of the Evaluation Setup in Experiments**
> > >
> > > The model and datasets used in our adaptive attack experiment are consistent with those in the main jailbreak evaluation. Specifically, we selected Qwen2-VL-7B-Instruct as the target model, conducted the attack on the full AdvBench dataset, and evaluated the results on a subset of 110 samples from HarmBench. In our previous experiments, we restricted the perturbation bound to 32/256.
> > >
> > > >**Q2. Stronger Adaptive Attack Evaluations**
> > >
> > > We referred to the example of designing adaptive attacks in [1]. In some cases, directly performing a defense-aware end-to-end attack **May not Yield the Best Results**, which is why we designed the random mask strategy. Of course, we are also very happy to conduct additional experiments as you requested as follows and we will include these experiments in the final version to improve the paper’s persuasiveness and completeness:
> > >
> > > **Strategy 1: Regularizing the Saliency Distribution.** Following the defense strategy outlined in GTM, we select the top-$K$ tokens that are originally targeted for masking. By applying a regularization penalty to these specific tokens, we force the perturbation to be distributed uniformly across the global tokens:
> > >
> > > $$\min_{\|\delta\|_\infty \le \epsilon}(L_C+ \lambda\sum g_i),$$
> > >
> > > where $L_C$ is the original CE loss, and $g_i$ represents the image token from the top-k.
> > >
> > >
> > > **Strategy 2: Defense-aware Gradient-based Attack.** We conducted a full-gradient attack against GTM. In each forward pass, we perform token masking exactly as dictated by the GTM defense mechanism. Subsequently, we optimize the perturbation using the standard cross-entropy loss. This strategy forces the attacker to establish a robust adversarial trajectory using only the sub-optimal, unmasked tokens, representing a highly aggressive "worst-case" adaptive evaluation.
> > >
> > > We fix the attack budget at 10,000 gradient steps. We set the perturbation bound to 32/256. For the hyperparameter $\lambda$ in Strategy 1, we conduct multiple trials and report the best-performing results. The detailed results are presented in the table below:
> > >
> > > |Perturbation Bound|W/O Defense|GTM with Stategie 1|GTM with Strategies2|
> > > |:---|:---|:---|:---|
> > > |32/256|82.73|4.55|19.09|
> > > |64/256|68.18|26.36|31.82|
> > >
> > > Additionally, we explored **Strategy 3: Differentiable Soft Masking**. We make the top-$K$ operation in the GTM mechanism differentiable via a sigmoid-based soft mask to integrate it into the backpropagation pipeline. However, the loss showed no significant decrease even after 10,000 iterations. This is likely attributable to severe gradient conflicts between the soft mask and the intrinsic cross-entropy loss [1], which ultimately led to the failure of the attack.
> > >
> > > In summary, we sincerely thank the reviewer for prompting us to conduct a more rigorous adaptive evaluation. Inspired by your insightful feedback, we formulated and evaluated three distinct defense-aware attack strategies: saliency regularization (Strategy 1), defense-simulated masking (Strategy 2), and differentiable top-$K$ approximation (Strategy 3). The comprehensive results demonstrate that even when facing a fully transparent adaptive attacker with a large perturbation budget in a worst-case scenario, GTM still retains a certain level of defensive capability. We will devote sufficient space in the final version to comprehensively discuss these adaptive attacks.
> > >
> > > [1] F., Tramer et al. (2020). On Adaptive Attacks to Adversarial Example Defenses. In Advances in Neural Information Processing Systems (NeurIPS).

---

### Official Review · Reviewer_ui3v · 2026-03-12

**Soundness:** 3
**Presentation:** 2
**Significance:** 2
**Originality:** 2
**Overall Recommendation:** 4
**Confidence:** 4

**Summary:**

This paper studies the impact of visual tokens in visual prompt injection attacks. Specifically, the authors show that successful attacks often rely on only a small subset of visual embedding tokens rather than the entire image representation. Based on this observation, they propose a gradient-based attribution method to identify these critical tokens and mask them during inference. Experiments across multiple LVLMs show that the proposed method can reduces attack success rates while maintaining the models’ normal performance.

**Compliance With Llm Reviewing Policy:**

Affirmed.

**Final Justification:**

I have increased my score, as the author’s follow-up rebuttal addressed my question.

**Key Questions For Authors:**

1. How robust is the method against adaptive attacks that distribute malicious signals across many tokens?
2. How stable are the identified critical tokens across different prompts?
3. Does the proposed defense generalize to closed-source LVLMs?

**Limitations:**

Yes

**Strengths And Weaknesses:**

Strengths:
1. The paper provides an insightful analysis of visual prompt injection attacks in LVLMs.
2. The proposed defense is simple and easy to implement.
3. The method is evaluated across multiple attack settings, demonstrating consistent reductions in attack success rates.

Weaknesses:
1. The proposed defense relies on the assumption that adversarial influence is concentrated in a small subset of visual tokens. However, adaptive attackers could potentially distribute malicious signals across many tokens, which may weaken the effectiveness of the masking strategy.
2. Gradients from deep networks are known to be noisy. Thus the authors should analyze the robustness of the proposed attribution method, for example by examining whether the identified tokens remain consistent across different text inputs.
3. The evaluation is limited to the open-source LVLMs. Evaluating the proposed defense on close-source LVLMs would show the generalizability of this proposed defense.

---

> ### Author Rebuttal · Authors · 2026-03-31
>
> Thank you for your careful review and valuable feedback. We have made every effort to address your concerns.
> >**W1&Q1. Robustness to Adaptive Attack**
>
> Evaluating robustness under adaptive and worst-case attackers is indeed crucial, and we agree with the reviewer that an adaptive adversary may attempt to disperse adversarial signals to bypass our masking strategy. To rigorously assess the robustness of GTM, we design an adaptive attack based on **random masking**, where 50% of the image embedding tokens are stochastically masked during optimization.  This encourages the attacker to learn redundant and spatially distributed adversarial features. Importantly, to properly optimize under such stochastic transformations, we adopt an **Expectation over Transformation (EOT)**[1] strategy during attack generation, which is a widely recognized adaptive attack method. We trained for 10,000 epochs. The results are summarized below:
>
> |                    | Without Masking | Random Masking       |
> | :----------------- | :------------------: | :------------------: |
> | **GTM** |         0.00         |         4.55         |
>
> As shown, even under this strengthened adaptive attack with EOT optimization, the attack fails to effectively bypass our defense. This demonstrates that GTM maintains strong robustness against adaptive strategies that attempt to distribute adversarial perturbations across tokens.
>
> Should the reviewer have any further suggestions regarding the design of the adaptive attack, please let us know. We are fully committed to incorporating additional experiments to address your concerns.
>
>
> > **W2\&Q2. Stability of Token Attribution Across Different Text Inputs**
>
> For the same adversarial image (generated by ImageHijack), we visualized the token selection frequency in our defense method across various text inputs. The results, shown in https://anonymous.4open.science/r/ICML-F31882/f3.png, reveal a highly uneven distribution: although some tokens may be selected occasionally due to local noise or specific interactions between text and image, **a small subset of tokens is consistently selected with a high frequency across nearly all different text prompts**. The stability of these "high-frequency tokens" provides strong evidence that our method is capturing structured, underlying adversarial signals rather than random gradient noise. If attribution were dominated by noise, we would expect randomly dispersed selection across tokens. The observed pattern contradicts this, confirming the reliability of our approach.
>
> > **W3\&Q3. Generalization to Closed-source LVLMs**
>
> We appreciate this practical concern. GTM requires white-box access to hidden states and gradients, which closed-source APIs do not expose. However, we argue that GTM holds significant practical value in two key deployment scenarios:
> * **For open-source deployments.** The open-source VLM ecosystem (e.g., LLaVA, Qwen, Phi) is rapidly growing, with widespread adoption in privacy-sensitive and customizable applications. For these practitioners, GTM offers a **lightweight, training-free, plug-and-play** defense that adds minimal latency—a unique advantage over heavy purification methods.
> * **For model providers.** Companies deploying VLMs as services **internally have white-box access** to their systems. They can integrate GTM into backend inference pipelines as a **server-level safeguard** with negligible computational overhead.
>
> Although direct application to black-box APIs is not feasible with the current design, the adversarial inputs can be monitored by our defense using smaller surrogate models that mimic the target VLM's behavior.
> [1] A. Athalye et al.(2018) Obfuscated Gradients Give a False Sense of Security: Circumventing Defenses to Adversarial Examples. In International Conference on Machine Learning (ICML).

---

> > ### Author Rebuttal · Reviewer_ui3v · 2026-04-02
> >
> > Thanks for your response.
> >
> > Regarding the adaptive attack, it is unclear whether this strategy effectively leads to a distributed adversarial signal across tokens, rather than simply introducing randomness.
> >
> > For the Stability of Token Attribution Across Different Text Inputs, I am interested in a more direct evaluation under adaptive settings, for adversarial images generated via the adaptive attack, do the identified critical tokens remain consistent when evaluated with diverse text prompts?

---

> > > ### Author Response · Authors · 2026-04-05
> > >
> > > Thank you very much for your review and valuable feedback on our work.
> > > > **Q1: Stronger Adaptive Attack**
> > >
> > > We referred to the example of designing adaptive attacks in [1]. In some cases, directly performing a defense-aware end-to-end attack **May not Yield the Best Results**, which is why we designed the random mask strategy.  We also conduct additional experiments as follows and we will include these experiments in the final version to improve the paper’s persuasiveness and completeness:
> > >
> > > **Strategy 1: Regularizing the Saliency Distribution.** Following the defense strategy outlined in GTM, we select the top-$K$ tokens that are originally targeted for masking. By applying a regularization penalty to these specific tokens, we force the perturbation to be distributed uniformly across the global tokens:
> > >
> > > $$\min_{\|\delta\|_\infty \le \epsilon}(L_C+ \lambda\sum g_i),$$
> > >
> > > where $L_C$ is the original CE loss, and $g_i$ represents the image token from the top-k.
> > >
> > > **Strategy 2: Defense-aware Gradient-based Attack.** We conducted a full-gradient attack against GTM. In each forward pass, we perform token masking exactly as dictated by the GTM defense mechanism. Subsequently, we optimize the perturbation using the standard cross-entropy loss. This strategy forces the attacker to establish a robust adversarial trajectory using only the sub-optimal, unmasked tokens, representing a highly aggressive "worst-case" adaptive evaluation.
> > >
> > > We fix the attack budget at 10,000 gradient steps. We set the perturbation bound to 32/256. For the hyperparameter $\lambda$ in Strategy 1, we conduct multiple trials and report the best-performing results. The detailed results are presented in the table below:
> > >
> > > |Perturbation Bound|W/O Defense|GTM with Stategie 1|GTM with Strategies2|
> > > |:---|:---|:---|:---|
> > > |32/256|82.73|4.55|19.09|
> > > |64/256|68.18|26.36|31.82|
> > >
> > > Additionally, we explored **Strategy 3: Differentiable Soft Masking**. We make the top-$K$ operation in the GTM mechanism differentiable via a sigmoid-based soft mask to integrate it into the backpropagation pipeline. However, the loss showed no significant decrease even after 10,000 iterations. This is likely attributable to severe gradient conflicts between the soft mask and the intrinsic cross-entropy loss [1], which ultimately led to the failure of the attack.
> > >
> > > In summary, we sincerely thank the reviewer for prompting us to conduct a more rigorous adaptive evaluation. Inspired by your insightful feedback, we formulated and evaluated three distinct defense-aware attack strategies: saliency regularization (Strategy 1), defense-simulated masking (Strategy 2), and differentiable top-$K$ approximation (Strategy 3). The comprehensive results demonstrate that even when facing a fully transparent adaptive attacker with a large perturbation budget in a worst-case scenario, GTM still retains a certain level of defensive capability. We will devote sufficient space in the final version to a comprehensive discussion of these adaptive attacks.
> > >
> > > >**Q2. Stability of Token Attribution Across Different Text Inputs in Adaptive Attack**
> > >
> > > We visualized the consistency of our token selection in the adaptive attack in https://anonymous.4open.science/r/ICML-F31882/f4.png. The images used here are the adversarial images from Strategy 2 with a perturbation limit of 64/256. As shown in the figure, under the adaptive attack setting, the distribution characteristics of the tokens consistently selected by our defense method remain largely the same as those in the non-adaptive attack scenario. The major difference is the emergence of more tokens that the model stably selects. This demonstrates that while the adaptive attack does, to some extent, disperse the originally concentrated harmful semantics, our defense method can still achieve a certain level of defense.
> > >
> > > [1] F., Tramer et al. (2020). On Adaptive Attacks to Adversarial Example Defenses. In Advances in Neural Information Processing Systems (NeurIPS).

---

### Official Review · Reviewer_fhVw · 2026-03-13

**Soundness:** 3
**Presentation:** 3
**Significance:** 3
**Originality:** 3
**Overall Recommendation:** 5
**Confidence:** 3

**Summary:**

This paper finds that adversarial attacks on vision-language models rely disproportionately on a small number of tokens, which can be identified. The proposed method computes the gradient of the magnitude of the hidden state at the last layer in the first generation position, w.r.t the image token embeddings. The token embeddings with the largest gradients are set to zero and then the full sequence is generated.

There is a theoretical result that shows that, based on an assumption, the tokens discovered via this method are approximately the same as the tokens which have the greatest impact on the full output, rather than just the first token generation. Empirically, this approach provides strong robustness with minimal impact on capability.

**Compliance With Llm Reviewing Policy:**

Affirmed.

**Final Justification:**

The author rebuttal has helped maintain my confidence in the my original review. I think the contribution, even without the additional adaptive attack experiments, is solid. I hope the authors implement the framing changes and include the needle-in-the-haystack experiments, and I maintain my original positive score.

**Key Questions For Authors:**

See above. Additionally, the finding that a small number of tokens contribute to adversarial attacks seems to be presented as novel, but I think that this discovery is already shared in SafePTR, which is cited? Can you clarify whether this is novel?

**Limitations:**

See above. The approach negatively impacts capability.

**Strengths And Weaknesses:**

Strengths:

- The paper provides a detailed explanation of the method as well as a theoretical analysis
- Extensive evaluations of the robustness of the method across multiple models are provided, as well as general capability evaluations

Weaknesses:

- While the impact is small, this approach hurts general capability.
- I find it kind of confusing that the identified tokens are consistently framed as adversarial tokens which contribute the most to the adversarial generation when, in fact, these are simply the tokens that contribute the most to the generation, whether they are adversarial or not, no? In effect, the approach masks the tokens that have the greatest impact on the generation, and this happens as well when none of them are adversarial; and it conveniently turns out that benign images tend to have more diffuse gradient magnitudes, so it doesn't harm performance terribly to remove the tokens that score the highest via the salience metric? I think this should be clarified.
- In a similar line to the above, it seems plausible that there are cases where a small number of token embeddings do contribute disproportionately to the generated output (e.g. needle-in-the-haystack cases). I would be interested in seeing the authors analyze how their method might impact such cases.

---

> ### Author Rebuttal · Authors · 2026-03-31
>
> Thank you for your valuable feedback, which will enhance the integrity of our paper.
> >**W1. The Loss of General Capability**
>
> We agree that masking tokens inherently removes some visual information, representing a fundamental safety-utility trade-off. However, as shown in our MM-Vet evaluation , our method maintains performance very close to the undefended baseline. We believe it offers a sophisticated trade-off between security and general capability when compared to existing robust defenses.
>
> >**W2. Why Masking Preserves Performance on Benign Tasks**
>
> We thank the reviewer for raising this important point. This observation indeed provides valuable insight into the strong performance retention of our method.
>
> To illustrate this, we have visualized the gradients for both benign and prompt-injected images (https://anonymous.4open.science/r/ICML-F31882/f2.png). As shown, the **gradient distribution for benign images is notably diffuse**, whereas **gradient distribution for prompt-injected images is highly concentrated**. This observation demonstrates that our method is capable of precisely masking a small subset of adversarial tokens, with minimal impact on normal task performance.
>
> We will incorporate this clarification into the final version of the paper to enhance its comprehensiveness.
>
> >**W3. Experiments for Needle-in-the-haystack**
>
> To address the concern regarding "needle-in-the-haystack" scenarios, we conducted additional experiments on **HR-Bench-4K**. HR-Bench-4K is a highly challenging dataset dedicated to tiny object recognition. It requires LVLMs to identify extremely small objects within 4K resolution (4032x4032) images. We evaluated our method on Qwen2-VL-7B using this dataset. The table below presents the accuracy on HR-Bench-4K alongside the standard MM-Vet benchmark scores for comparison:
>
> |   | HR-Bench-4K Accuracy (%) | MM-Vet Score |
> | :--- | :--- | :--- |
> | **W/O Defense** | 67.62 | 62.3 |
> | **Ours (GTM)** | 61.75 | 56.8 |
>
> As shown in the table, our method causes only a slight performance drop on HR-Bench-4K compared to the undefended baseline. Crucially, the magnitude of this drop is consistent with the minimal decrease observed on standard tasks like MM-Vet. This finding suggests a key distinction between natural visual features and adversarial perturbations. Even for naturally occurring tiny objects, the corresponding token gradients remain relatively dispersed across the image embeddings, and they do not exhibit the extreme gradient concentration seen in prompt injection triggers. These results demonstrate that our proposed defense remains robust and applicable, even in highly sensitive visual recognition scenarios.
>
> >**Q1. The Novelty of Token Sparsity**
>
> Compared to the token sparsity discussed in SafePTR, the key differences in our findings are:
>
> 1. SafePTR aims primarily to locate and intervene on "**toxicity switches**" within the model, whereas our approach focuses on uncovering the latent "**harmful semantics**" concealed within the image embedding tokens. While the safety tokens identified by SafePTR rely on the model's inherent alignment to achieve defense, our method directly localizes and neutralizes the harmful semantics in the visual input, thereby blocking the adversarial manipulation at its source **without depending on the model's internal safety mechanisms**.
> 2. SafePTR identifies the sparsity of safety tokens within jailbreak attacks (with its experiments limited to **typography-based jailbreaks**), whereas we uncover the sparsity of adversarial tokens across a much broader class of **general perturbation-based prompt injection attacks** (including jailbreak attack).
>
> We will cite SafePTR in the final version of the paper and include a discussion on the differences between the two.

---

> > ### Author Rebuttal · Reviewer_fhVw · 2026-04-03
> >
> > I thank the reviewers for the detailed rebuttal.
> >
> > W2. I think the framing here could still be improved, and is perhaps slightly misleading. In any input, the tokens that are masked are those that contribute the most to the generation, correct? In this case, even if the distribution for prompt-injected images is concentrated, it seems incorrect to frame the masked tokens as always being "adversarial."
> >
> > W3. I appreciate the needle-in-the-haystack experiments. The performance drop appears to be moderate. It would be helpful if the authors could include confidence intervals for the evaluation scores.

---

> > > ### Author Response · Authors · 2026-04-05
> > >
> > > Thank you very much for your recognition of our work and your valuable suggestions.
> > > >**Q1. Terminology Revision for Masked Tokens**
> > >
> > > Thank you for pointing this out. We will revise the text following your suggestion and avoid referring to the masked tokens uniformly as “adversarial tokens,” even in the prompt injection setting. The revision will be roughly as follows:
> > >
> > > Our method identifies and masks the tokens that contribute most to the current generation.
> > > Based on this clarification, we will further explain the distinction between benign and adversarial settings in terms of the distribution of these high-contribution tokens. For benign images, the gradient is typically more diffuse, so masking a small number of top-ranked tokens tends to have only limited impact on generation quality. In contrast, for prompt-injected images, the gradient distribution for prompt-injected images is highly concentrated, so masking these tokens is more likely to effectively suppress the injected behavior.
> > >
> > > >**Q2. Confidence Intervals for Needle-in-the-haystack Experiments**
> > >
> > > We thank the reviewer for appreciating our experiments and for the constructive feedback.  We agree that providing confidence intervals makes the evaluation of the performance drop more rigorous. Following your advice, we have repeated the experiments for each setting 5 times using different random seeds. The confidence intervals results are presented in the table below:
> > >
> > > | Method | HR-Bench-4K Accuracy (%) | MM-Vet Score |
> > > | :--- | :--- | :--- |
> > > | **W/O Defense** | $67.62 \pm \text{0.00}$ | $62.3 \pm \text{0.20}$ |
> > > | **Ours (GTM)** | $61.80  \pm \text{0.09}$ | $56.76 \pm \text{0.19}$ |
> > >
> > > We will update the final version of the manuscript to include these confidence intervals in the main results.

---

### Decision · Program_Chairs · 2026-04-30

**Decision:**

Accept (regular)

**Comment:**

This paper proposes a lightweight defense (GTM) that localizes and masks critical visual tokens to mitigate prompt injection attacks in vision-language models. Reviewers generally agree the method is simple, efficient, and empirically effective, with solid theoretical grounding. The main concerns focus on adaptive attack robustness, potential impact on general capability, and evaluation completeness. The authors provided additional experiments and clarifications in the rebuttal, which addressed most concerns, though some questions on adaptive attacks remain partially unresolved. Overall, reviewer scores remain positive.

After reading the paper and rebuttal, the AC recommends acceptance of the paper.